# Alternative Splicing in the Regulatory Circuit of Plant Temperature Response

**DOI:** 10.3390/ijms24043878

**Published:** 2023-02-15

**Authors:** Rong Xue, Ruirui Mo, Dongkai Cui, Wencong Cheng, Haoyu Wang, Jinxia Qin, Zhenshan Liu

**Affiliations:** 1State Key Laboratory of Crop Stress Biology for Arid Areas, College of Agronomy, Northwest A&F University, Xi’an 712100, China; 2State Key Laboratory of Crop Stress Biology for Arid Areas, Institute of Future Agriculture, Northwest A&F University, Xi’an 712100, China

**Keywords:** splicing isoforms, temperature change, regulatory network, heat stress, cold stress

## Abstract

As sessile organisms, plants have evolved complex mechanisms to rapidly respond to ever-changing ambient temperatures. Temperature response in plants is modulated by a multilayer regulatory network, including transcriptional and post-transcriptional regulations. Alternative splicing (AS) is an essential post-transcriptional regulatory mechanism. Extensive studies have confirmed its key role in plant temperature response, from adjustment to diurnal and seasonal temperature changes to response to extreme temperatures, which has been well documented by previous reviews. As a key node in the temperature response regulatory network, AS can be modulated by various upstream regulations, such as chromatin modification, transcription rate, RNA binding proteins, RNA structure and RNA modifications. Meanwhile, a number of downstream mechanisms are affected by AS, such as nonsense-mediated mRNA decay (NMD) pathway, translation efficiency and production of different protein variants. In this review, we focus on the links between splicing regulation and other mechanisms in plant temperature response. Recent advances regarding how AS is regulated and the following consequences in gene functional modulation in plant temperature response will be discussed. Substantial evidence suggests that a multilayer regulatory network integrating AS in plant temperature response has been unveiled.

## 1. Introduction

In eukaryote, multilayer regulations of gene function have evolved to couple with their functional complexity, where splicing is an essential step in gene expression and functional regulation. The majority of eukaryotic pre-mRNAs contain noncoding introns, which need to be removed to produce mature mRNAs [1,2]. This pivotal process is called mRNA splicing and is achieved by a large ribonucleoprotein complex, the spliceosome [2,3]. Different mature transcripts can be produced from the same pre-mRNA by alternative splice sites selection, which is a ubiquitous phenomenon in eukaryotes called alternative splicing (AS) [4,5,6]. At least 61% of intron-containing genes in the plant model species *Arabidopsis thaliana*, 60% in *Drosophila melanogaster* and more than 95% in humans were reported to be alternatively spliced, greatly increasing the complexity of transcriptome and proteome [5,7,8].

A complex regulatory network integrating AS has been largely established from enormous studies in animal systems [2,9]. AS can be regulated at different levels, including chromatin modification, transcription elongation, RNA binding proteins, RNA modification, RNA structure and small RNAs [2,9]. Meanwhile, AS itself can modulate gene function through different mechanisms, such as coupling to nonsense-mediated mRNA decay (NMD) pathway to modulate mRNA expression, encoding proteins with distinct functions or localization, encoding truncated proteins with dominant negative effect, or modulating miRNA generation [4,10,11,12,13,14,15].

As a sessile organism, it is critical for plants to properly respond to ambient temperature changes, including seasonal and diurnal temperature variation and extreme temperature stresses. Temperature is one of the major environmental factors affecting crop growth, yield and distribution [16,17,18]. Extreme temperature weathers, including high temperature or heat stress, low temperature or chilling stress and freezing temperature/stress, are major threats to global food security [16,18,19]. As a consequence of global warming, heat stress in particular is affecting agricultural crops more frequently and more severely [18]. High temperature usually has negative effects on crop physiology (e.g. decreased photosynthesis), plant growth, root development, flowering and grain filling, resulting in disrupted pollination, flowering and crop yield and quality [16,17,18]. Knowledge about how plants adapt, respond and tolerate extreme temperature stresses is vital for the enhancement of crop productivity under changing climatic conditions.

AS is reported to play key roles in plant temperature response, which has been well documented in many previous reviews [1,20,21,22,23,24,25]. For example, many important regulatory genes, such as flowering genes *FLM* and *FLC* in seasonal temperature sensing [26,27,28,29,30,31], *CCA1* in diurnal temperature response [32,33] and heat stress transcription factor genes (*HSFs*) in temperature stress response [34,35,36], were reported to undergo AS regulation. Genome-wide studies also demonstrated that hundreds to thousands of genes displayed temperature-induced alternative splicing in different plant species [37,38,39,40,41,42,43]. Plants respond to temperature changes in a systematic manner involving regulation mechanisms from transcriptional levels to post-translational levels [44,45,46]. Many recent studies have focused on the links between AS and other regulatory mechanisms, which leads to the emerging of a multilayer regulatory network in plant temperature response. Here, we reviewed studies in plant temperature responsive AS and highlighted the relationship between AS and other mechanisms in forming a complex temperature regulatory network.

## 2. Pre-mRNA Splicing Process

Correct splicing of pre-mRNAs is crucial for eukaryotic organisms [47]. In animals and plants, most of the introns (>99%) are U2-type, which contain the canonical GU–AG dinucleotide intron boundaries (Figure 1a) [47]. Three sequence elements in introns are essential for splicing: the 5${}^{\prime }$ splice site (with a conserved GU), the 3${}^{\prime }$ splice site (with a conserved AG) and the branch point (with a conserved A), which is about 20–50 nt preceding the 3${}^{\prime }$ splice site (Figure 1a) [2,47]. A polypyrimidine tract (rich in pyrimidines, especially uridines) of 15–20 nucleotides long usually lies between the branch point and the 3${}^{\prime }$ splice site to facilitate the recognition of the 3${}^{\prime }$ splice site (Figure 1a) [2,47].

From a pre-mRNA′ perspective, the splicing process is simply accomplished by two transesterification reactions [2,47]. In the first transesterification reaction, the 5${}^{\prime }$ end of the intron is cleaved from the 5${}^{\prime }$ exon and is ligated to the “A” nucleotide at the branch point to form a lariat structure in the intron (Figure 1a) [47]. In the second transesterification, the 3${}^{\prime }$ end of intron is cleaved from the 3${}^{\prime }$ exon to release the intron lariat structure, and the 5${}^{\prime }$ exon and the 3${}^{\prime }$ exon are ligated together (Figure 1a) [47]. These two transesterification reactions are catalysed by a large ribonucleoprotein complex, the spliceosome. From a spliceosome’ perspective, the splicing process is complex, involving the interaction of five snRNAs and about a hundred proteins [2,3,48,49]. The spliceosome is a multicomponent ribonucleoprotein (RNP) complex, the assembly of which results from the dynamic organization of five small nuclear RNPs (U1, U2, U4, U5 and U6 snRNPs) and many associated proteins in a step-wise manner [3,48,49]. A spliceosome is a ribozyme with a catalytic center composed of RNAs (snRNAs) [3,48]. Proteins in spliceosomes function as backbones to ensure juxtaposition of the 5${}^{\prime }$ splice site and 3${}^{\prime }$ splice site in the catalytic center [3,48]. The current model of how spliceosome catalyses splicing is from studies in humans and yeast [3,50,51,52]. Simply said, the ATP-required pre-mRNA splicing is executed with the following steps: (1) U1 snRNP firstly recognizes the 5${}^{\prime }$ splice site by base-pairing between the 5${}^{\prime }$ end of U1 snRNA and sequences around the 5${}^{\prime }$ splice site (Figure 1b) [53]; (2) U2 snRNP associates with the branch point to form the spliceosome precursor (Figure 1b) [51]; (3) U4/U5/U6 snRNPs interact with the spliceosome precursor to complete spliceosome assembly [3,43]; (4) U1 and U4 snRNPs are released and other protein factors are recruited to form a catalytically active spliceosome [3,49]; and (5) transesterification reactions are accomplished and snRNPs are released [3,49]. Spliceosome assembly needs to occur repeatedly every time an intron is removed from a pre-mRNA in a eukaryotic nucleus.The sequences around the splice site and branch point are loosely conserved in plants and animals [1,11,43]. Multiple splice sites can coexist on pre-mRNA to compete with each other to form alternative splice sites, which is the basis of alternative splicing, a key node in the plant temperature regulatory network (Figure 2).

## 3. Upstream Mechanisms Regulating AS in Plant Temperature Response

### 3.1. Epigenetic Control of Temperature-Responsive AS

Chromatin status (DNA methylation, histone modification and nucleosome occupancy) is usually considered as being related to transcriptional regulation. Furthermore, many recent studies in animal systems have shown that epigenetic mechanisms can directly affect gene splicing [54,55,56]. The structural units of chromatin are formed by nucleosomes, and Nucleosome localization on DNA sequences was found to contribute to exon selection [54,55,56]. Variations in nucleosome occupancy are associated to genome-wide alterations in pre-mRNA splicing [57,58]. Genome-wide mapping of nucleosome locations revealed that nucleosomes occupy exons more frequently than introns [59]. Interestingly, stable nucleosome occupancy has been reported to be stronger in exons with weak splice sites [60]. Histone modifications also contribute to exon recognition. Many histone modifications, such as H3K36me3, H2BK5me1, and H3K27me1/2/3, are more abundant over exons versus introns [61,62]. Histone modifications can also directly impact splicing regulation by recruiting splicing factors through chromatin-binding proteins [63]. Similar to histone modification, DNA methylation was also suggested to impact global exon–intron recognition and splicing of a subset of genes directly [64,65]. DNA methylation occurs with higher frequency in exons than in introns over the whole genome [64]. Exons with lower expression levels also have lower DNA methylation levels than highly expressed exons [64,65]. In plants, several studies have also revealed the epigenetic control of alternative splicing [56,66,67,68,69]. For example, CG-type DNA methylation level in rice was found to be markedly higher in exons than in the adjacent introns [70]. DNA methylation loss in a null mutant of a major CG methyltransferase in rice (*OsMet1-2*) was significantly associated with AS pattern changes [70].

It has long been known that temperature changes can lead to genome-wide epigenetic alterations [71,72,73,74,75]. For example, cold stress can globally enhance chromatin accessibility and H3K4me3-H3K27me3 modification levels in the genic region of cold-responsive genes in potato [76]. Several studies have observed the whole-genome-scale association between epigenetic change and AS alteration in plant temperature responses [67,68,69]. A genome-wide study using ChIP-seq and RNA-seq in Arabidopsis showed that temperature-induced AS (from 16 °C to 25 °C) was strongly enriched in genes with higher H3K36me3 modification, including flowering time regulators *FLM*, *MAF2*, and *FCA*, as well as circadian clock regulators *PRR3* and *PRR7* [68]. Indeed, mutations in H3K36me3 writers, eraser, or readers affected high-temperature-induced flowering, suggesting the roles of histone modification in flowering control via modulating AS [68]. By using RNA-Seq, MNase-Seq and whole-genome bisulfite sequencing, two recent studies found that DNA methylation and nucleosome occupancy splice sites, and alternative exons/introns were strongly associated with cold-induced AS in Arabidopsis [67,69].

Co-transcriptional splicing provided a mechanistic basis for the cis-regulatory effect of chromatin status on AS, as it links splicing and chromatin spatially and temporally [9,14,77,78]. Co-transcriptional splicing was firstly observed in yeast and animals, and was also revealed in plants recently [9,14,77]. A study of chromatin-associated RNAs in Arabidopsis indicated that co-transcriptional splicing is a genome-wide phenomenon correlated with the splicing outcome of mature mRNAs [79]. Co-transcriptional efficiency was also found correlated with chromatin modification, transcription and the number of introns and exons [80]. Intriguingly, the process of transcription also influences AS [77,81,82]. In Arabidopsis, native elongating transcript sequencing (NET-seq) showed that the phosphorylation of Pol II CTD domain facilitates the interaction between Pol II and spliceosome, influencing alternative splicing [77,81]. However, the detailed mechanisms of how chromatin status regulates temperature-responsive AS are still largely unknown.

### 3.2. Splicing Factors and Spliceosome Components Modulate Temperature-Responsive AS

Splicing factors (SFs) refer to RNA-binding proteins which associate with pre-mRNA and core spliceosome in constitutive splicing or alternative splicing [20,22,83]. As *trans* factors, splicing factors were reported to promote or repress splicing by interacting with *cis*-elements (sequence motif such as splicing enhancer or splicing silencer) on pre-mRNA [83]. Splicing factors were documented to be widely involved in plant development and response to environmental stimuli [1,20,43,83]. Serine/arginine-rich (SR) proteins are the most widely studied splicing factors in plants. A typical SR protein consists of one or two RNA recognition motifs (RRM) and an arginine/serine-rich domain (RS) [83]. The RRM motifs are involved in binding to *cis*-elements on pre-mRNA to promote constitutive splicing or mediate alternative splicing, while the RS domain mediates protein–protein interactions to activate splicing [83,84]. It has been reported that SR proteins can bind exonic splicing enhancers and activate splicing by recruiting core splicing factors such as U1 snRNP or U2AF [83,85]. SR proteins are highly conserved in plants and metazoan [83]. Plants generally possess more SR family members than animals and other eukaryotes [83]. For example, there are 18 SR genes in Arabidopsis, 22 in rice and 25 in soybean, while only 12 in humans [83,86,87,88]. The expansion of the SR family in plants was considered to have resulted from polyploidization events during evolution and is suggested to be related to environmental adaptation [83,89].

Several studies have revealed the important roles of splicing factors in plant temperature response [23,90]. For example, AtSF1 was suggested to regulate plant flowering and heat shock response [34,35,36]. AtSF1 is responsible for the ambient-temperature-dependent AS of flowering gene *FLM*, contributing to temperature-responsive flowering in Arabidopsis [91]. In addition, Arabidopsis *sf1* mutants are hypersensitive to heat stress and displayed impaired splicing pattern of *HSFA2* gene and altered expression of HSFA2 targets [34,35,36]. However, direct evidence supporting the interaction between AtSF1 and its target mRNAs was lacking in these studies. Another splicing factor in Arabidopsis, SR45, was reported to bind specifically to the pre-mRNA of circadian clock gene *CCA1* [92]. SR45 mutation or overexpression influenced the splicing pattern of *CCA1* in response to high temperature, suggesting the potential role of SR45 in diurnal temperature-dependant splicing of *CCA1* [92].

Spliceosome components and associated proteins were also involved in modulating AS in plant temperature response. The protein core of spliceosome is composed of Sm ring proteins and the related Sm-like complex [84,93]. Sm protein E1 (SME1), an Arabidopsis homolog of the SME subunit of Sm ring, can control spliceosome activity and specificity to different pre-mRNAs and negatively regulate cold adaptation in Arabidopsis [94]. In contrast, an Sm-like protein, PCP, was found to positively regulate development in response to low temperature through modulating AS in Arabidopsis [95]. *PCP* mutation caused striking development deficit, specifically under low temperature [95]. Additionally, spliceosome assembly factor GEMIN2 was found to control the AS of several circadian clock genes and attenuate the effects of temperature on the circadian period in Arabidopsis [96]. RBP45d, an RNA-binding protein in the U1 snRNP complex, directly regulates alternative splicing of a specific set of genes, including *FLM*, in response to low temperature [97]. STABILIZED 1 (STA1), a U5-snRNP-associated protein homolog, was reported to be involved in cold and heat stress tolerance in Arabidopsis [98,99]. *Sta1* mutant plants showed defects in the splicing of temperature-responsive genes *COR15A*, *HSFA3* and *HSA32* [98,99].

Splicing factors and spliceosome components can be regulated by temperature changes via different mechanisms [1,20,43,100]. For example, SR proteins were reported to be modulated at transcriptional, post-transcriptional and protein phosphorylation levels [83]. Two heat shock transcription factors, HSF-A1a and HSF-A2, can promote the expression of *SR* genes in tomato [101]. A CLK kinase is identified as a thermosensor to control global splicing via modulating the phosphorylation status of SR proteins in mammals [102]. Moreover, *SR* genes were also reported to undergo AS in response to low temperature [103]. Many SR proteins can regulate the AS of their own pre-mRNA, forming a negative feedback loop to maintain their expression [103]. In addition, Arabidopsis CDKG1 and CDKG2 can modulate the AS of U2AF65A, a key component in U2-snRNP, in response to temperature change [104].

## 4. Downstream Pathways Modulated by AS in Plant Temperature Response

### 4.1. Alternative Splicing Modulates mRNA Expression and Localization in Response to Temperature Changes

AS can generate transcript isoforms with different stability, localization or polyadenylation sites, and thus leads to different functionality [1,4,13,105,106]. Many studies have shown that AS frequently generates transcripts harboring premature termination codon (PTC), which can be degraded by the nonsense-mediated decay (NMD) pathway [5,32,107]. Therefore, AS coupling to the NMD pathway can modulate the abundance of functional transcripts [108]. For example, in Arabidopsis, heat shock factor *HSF-A2* was reported to give rise to different splicing isoforms depending on environmental temperature [90]. A heat-induced AS event generates PTC containing AS isoform *HsfA2-II*, which can be degraded through NMD [109]. AS–NMD coupling is also involved in circadian clock regulation under temperature stresses. Heat stress induces intron retention in *ELF3* (*EARLY FLOWERING 3*), which is part of the evening complex, leading to transcripts that are targeted by NMD [22]. Cold stress can stimulate AS of the *LHY* (*LATE ELONGATED HYPOCOTYL*) gene, resulting in a nonfunctional *LHY* transcript that contains a PTC and is degraded by NMD [110].

In plants, intron retention is the most frequent AS event and usually leads to PTC-harboring transcripts, but only a subset of them can be degraded by the NMD pathway [5,111,112,113]. This is partly explained by the nucleus sequestration of intron-retention transcripts [90,114,115]. For many genes in Arabidopsis, eg., *SR30*, *RS2Z33* and the *SEF* factor, the intron-containing transcripts were found predominantly sequestered in the nucleus to escape the NMD pathway [116,117]. Emerging evidence has shown that the removal of specific introns appeared to be a “check point” for nuclear export of mRNA responding to environmental stimuli [82,118]. In Arabidopsis, the U1 snRNP subunit LUC7 was found to regulate nuclear sequestration of intron-containing transcripts of stress-responsive genes, particularly under cold stress conditions [118]. A recent study applying nanopore full-length RNA sequencing to chromatin-bound RNA in Arabidopsis revealed that post-transcriptional splicing rather than co-transcriptional splicing contributes to the nuclear sequestration of intron-containing transcripts [82]. These transcripts were further found to be enriched in stress-responsive pathways, especially in cold and heat stress responses [82]. Thus, intron retention triggered nuclear sequestration of unspliced transcripts, which may serve as a new regulatory mechanism to fine-tune gene expression in plant stress response.

### 4.2. Effects of Alternative Splicing on Translational Regulation in Plant Temperature Response

A major effect of AS is that it dramatically increases the complexity of proteome by producing transcripts with different protein-coding potential [119]. Indeed, AS in humans is reported to expand the proteome by at least a factor of five compared with the number of protein-coding genes [120], while in plants the effects of AS have not yet been widely addressed at the proteome level. Several studies have shown that AS transcripts can encode proteins with distinct functions in plant temperature response [20]. A common mechanism is that the dominant negative effects that protein variants with impaired domain structure caused by AS can compete with the “normal” protein to inhibit its function [22]. For example, the Arabidopsis *FLM* (*FLOWERING LOCUS M*) gene is subject to temperature-dependent AS and produces two protein-coding splice variants, *FLM-β* and *FLM-δ*. These two AS transcripts can be translated into proteins with opposing functions, acting as a repressor (*FLM-β*) and promoter (*FLM-δ*) of flowering [26,27,30,121,122]. At low temperature, *FLM-β* is dominantly expressed to form a protein complex with transcription factor SVP and represses flowering [27]. At elevated temperature, FLM-δ is upregulated to competitively interact with SVP. However, the resulting SVP-FLM-δ complex is unable to bind DNA, releasing the floral promoters of *FT* and *SOC1* genes from repression and thereby promoting flowering [27]. A dominant negative effect induced by AS was also found in temperature-dependent circadian control. Under heat stress, the AS of the circadian clock regulator CCA1 pre-mRNA generated a truncated protein lacking DNA-binding domain, which can compete with the full-length CCA1 for interaction partners, generating nonfunctional protein complexes [33,123]. In contrast, cold temperatures result in a strong downregulation of the truncated CCA1 protein via AS to enhance the function of the full-length CCA1 protein [32,110].

A dominant negative effect was also reported to modulate plant heat stress response. Prolonged heat stress in lily can induce the production of an AS isoform *HSF-A3B-III*, which encoded a truncated protein interacting with full-length HSFA3 to reduce the adverse effects of excessive HSFA3 accumulation, enhancing tolerance to prolonged heat stress in lily [124]. In Arabidopsis, the AS of the *SGR5* gene produces two protein variants, full-length SGR5α and truncated SGR5β [125]. High temperature can promote the generation of SGR5β to form nonfunctional heterodimer with SGR5α and modulate the gravitropic response of inflorescence stems [125].

A dominant negative effect only offers more regulatory flexibility to fine-tune gene functionality. AS can also generate protein variants with distinct functions contributing to functional diversification [119]. An interesting example is how a typical heat sensor, ion channel TRPV1, is tuned into a more sensitive infrared detector via AS in vampire bats [126]. The TRPV1 ion channel is a vertebrate thermal sensor that can be activated by noxious heat (>43 °C) in somatic afferents. In vampire bats, the AS of the *TRPV1* gene produces a channel with a truncated carboxy-terminal cytoplasmic domain, which is hypersensitive to infrared and can be activated at as low as 30 °C [126]. This splicing event occurs exclusively in trigeminal ganglia surrounding the vampire bat’s nose to detect infrared radiation from warm-blooded prey [126]. However, no such contrasting cases have been reported in plant temperature sensing. Studies in plants also revealed the roles of AS in generating protein variants with diversified functions. For example, both *HSFA2* AS isoforms, *HSFA2-I* and *HSFA2-II*, encode proteins with similar transcriptional activation activity under heat stress [127]. However, the absence of NES (Nuclear Export Signal) in HSFA2-II facilitates its nuclear localization and thus confers higher activity in promoting *HSP* genes transcription [127]. The Arabidopsis *SR45* gene possesses two splicing isoforms, *SR45.1* and *SR45.2*, which encode proteins differing by eight amino acids, including several putative phosphorylation sites [128,129]. Remarkably, *SR45.1* can rescue the floral but not the root phenotype in Arabidopsis *sr45* mutant, while *SR45.2* could complement the root growth defect [128].

### 4.3. Other Mechanisms Possibly Involved in AS Modulating Plant Temperature Response

Extensive studies revealed that AS can couple to various mechanisms to modulate gene function, which dramatically extend the AS regulatory network in eukaryotes. For example, AS can be coupled to miRNA-related regulation by either modulating pre-miRNA splicing to control miRNA production or producing AS isoforms of miRNA targets with different sensitivity to miRNA binding [130,131,132]. AS occurring at the 3’-end of pre-mRNA can generate transcripts with different poly(A) signal motifs to affect polyadenylation site selection, affecting mRNA stability [133]. AS occurring at the 5’-end of transcripts was reported to affect translation initiation efficiency [134,135].

Plant temperature response is a systematic reaction involving various molecular mechanisms. It is reasonable to speculate the coupling of AS to different mechanisms in plant temperature response. However, very few such studies have been reported. Interestingly, a recent study of the plant vernalization gene *TaVRN1* has revealed that lncRNA *VAS* is derived from the sense strand of *TaVRN1* by AS during vernalization [136]. *VAS* was physically associated with transcription factor TaRF2b to promote *TaVRN1* expression and flowering [136].

## 5. Concluding Remarks and Future Perspectives

To date, extensive studies have reached a solid conclusion that alternative spicing is an essential component in plant temperature response (Figure 2), from responding to daily and seasonal temperature changes to coping with extreme temperature stresses [23,33,34]. Splicing regulation is a node in the whole regulatory network of plant temperature response. Recent studies have revealed that almost all the upstream regulations, including chromatin modification, transcription rate, RNA binding proteins, RNA structure and RNA modifications, can affect splicing status [10,54,55,56]. Most studies focused on splicing factors (RNA binding proteins) in plant temperature response, while other mechanisms are beginning to be unveiled by recent advances [20,22,83]. Splicing status change holds the potential to affect a wide range of regulation mechanisms, such as the NMD pathway, translation efficiency, miRNA biogenesis and translation of protein variants [22,27,90,109,110,130,131,132,133,134,135]. The effect of splicing regulation by NMD pathway and dominant negative effect has been established in plant temperature response, while others are still to be explored [22,27,33,90,109,110,123].

By what mechanisms temperature signals are passed to splicing control remains unclear. It is well known that several transcription factors are master regulators in the transcriptional control of temperature response [137]. For example, transcription factor HSF serves as master regulator to modulate the transcription of a wide spectrum of heat-responsive genes in heat stress response [137,138]. Thus, an interesting question is whether any splicing factors serve as master regulators to modulate AS during plant temperature response. Emerging evidence has demonstrated that splicing factors, such as SR proteins or spliceosome components, are involved in this process, but their exact roles remain to be determined [20,22]. Epigenetic regulation might be another possible mechanism to sense temperature and subsequently modulate AS [73]. Current studies mostly focus on the genome-wide association between epigenetic changes and AS during plant temperature response [68]. The underlying mechanisms still require further exploration.

Intron retention is the most frequent AS event in plants and prevalently generates transcripts with PTC [90]. Theoretically, PTC-harboring transcripts can be translated into truncated proteins, which usually leads to inhibition of their full-length counterpart in a dominant negative manner. An interesting question is whether a dominant negative effect serves as a genome-wide common mechanism to achieve quick self-silencing of genes. However, we still know little about how much of PTC-harboring transcripts can be translated into truncated proteins in living cells, as current proteome technologies cannot properly distinguish truncated proteins from full-length ones [119]. Recent advances in protein sequencing using nanopore technologies pave a path for solving this problem [21]. Conclusively, alternative splicing control is an essential component in the regulatory network of plant temperature response, but the links between alternative splicing and other mechanisms are just emerging.

## Figures and Tables

**Figure 1 ijms-24-03878-f001:**
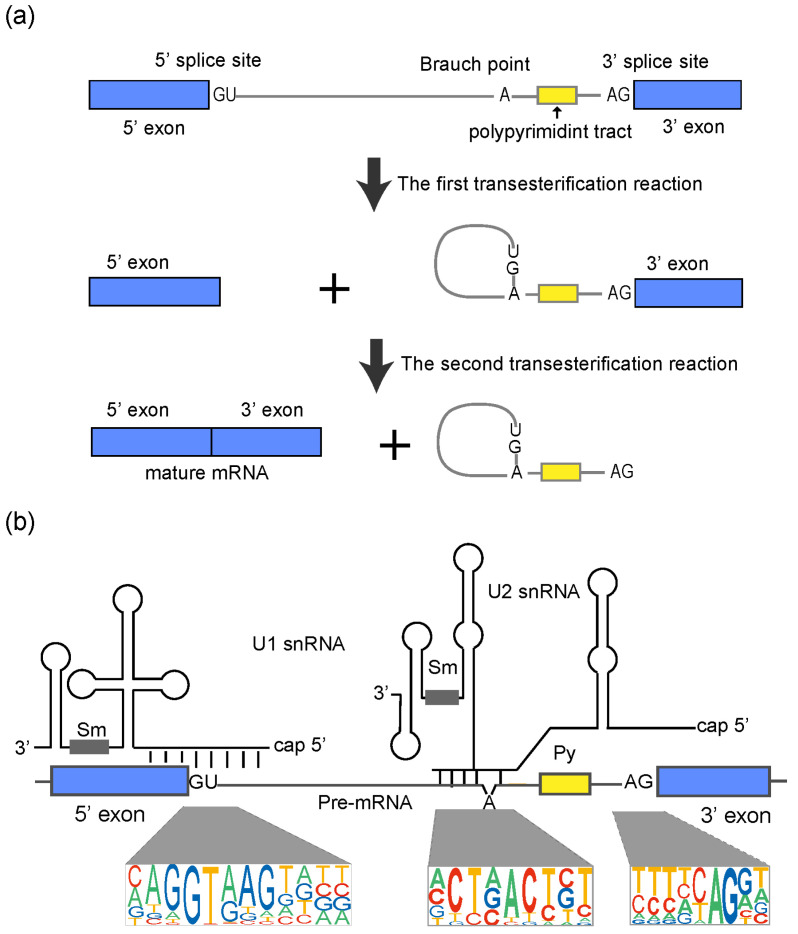
Process of splicing and the base-pairing between snRNAs and pre-mRNA during splice site recognition. (**a**) Essential sequence elements on pre-mRNA and how they participate in the two transesterification reactions to accomplish splicing. (**b**) Base-pairing between U1/U2 snRNAs and pre-mRNA during 5${}^{\prime }$ splice sites and 3${}^{\prime }$ splice sites recognition. The 5${}^{\prime }$ end of U1 snRNA base-pairs with sequences around the 5${}^{\prime }$ splice site. U2 snRNA base-pairs with nucleotides flanking the branch point and leaving the branch point “A” as a bulge. Sequences around the 5${}^{\prime }$ splice site and the branch point are loosely conserved, which affects the interaction between snRNAs and pre-mRNA, as well as the process of splicing. Graphical representations of the consensus sequences at both splice sites and the branch point are shown, in which the size of each letter represents the frequency of each base at each position over all introns in humans.

**Figure 2 ijms-24-03878-f002:**
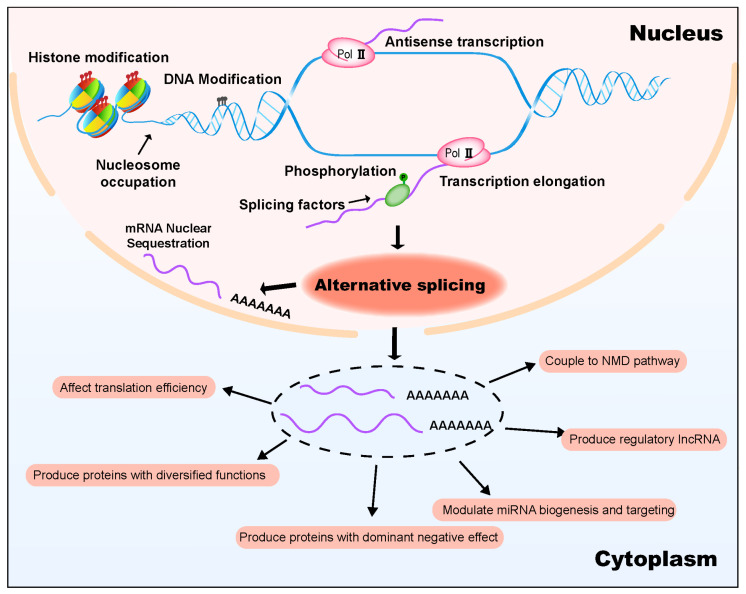
Alternative splicing participated regulatory circuit in plant temperature response. AS is a key node of the regulatory network in plant temperature response. AS can be modulated by various upstream regulatory mechanisms, such as DNA methylation, histone modification, nucleosome occupation, splicing factors and protein phosphorylation, as well as regulate gene function through different downstream mechanisms, such as coupling to NMD pathway to modulate mRNA expression abundance, encoding functionally distinct protein variants, producing regulatory lncRNAs, affecting miRNA-related regulation and regulating translation efficiency. This AS participated regulatory network in plant temperature response is summarized in detail in this review.

## Data Availability

Not applicable.

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
