# Peer review of "Alternative Splicing in the Regulatory Circuit of Plant Temperature Response"

_ijms, 2023, doi:10.3390/ijms24043878_

Round 1
Reviewer 1 Report
In this review, authors have presented the recent advances regarding how AS is regulated and the following consequences in gene functional modulation in plant temperature response will be discussed. Substantial evidences have suggested that a multi-layer regulatory network integrating AS in plant temperature response is unveiling. Overall, the authors have done a great job and prepared and nice and attractive article. Mainly, the introduction needs to be further improved by discussing the impacts of extreme temperature events on crop plants. Some other suggestions are:
Please use some diverse keywords which have not been used in the title.
In introduction, the authors should add a new detailed paragraph about how climate change and temperature stress impact crop production. I suggest checking and citing these articles such as doi: 10.1080/07388551.2021.1898332; 10.1080/21645698.2022.2106111.
Please go through the whole text and define the abbreviations on the first mention in the main text.
Make sure all gene names are in italics, but not the proteins etc.
Section 5, update the heading and add some concluding marks and then go on towards future recommendations.
Reviewer 2 Report
This submitted manuscript is written well, and the authors summarize the alternative splicing by temperature change in plants. However, I found some points to improve this manuscript. I hope those are helpful to bash up on this manuscript.
1. I found some similar reviews about alternative splicing and temperature. So, what is unique about the submitted manuscript? If authors could include this point in the introduction or elsewhere, it would help readers select this paper.
2. It would be helpful if the authors described a little more specific information about the temperature at some places in the main text (especially the third paragraph). For example, “temperature change” in line 192 is too abstract for readers to understand whether it is high or low temperature.
3. Line 9: Please add nonsense-mediated mRNA decay before NMD
4. Line 171: What SR stands for? Serine/arginine-rich?
5. Line 331 Erase the space in “reg ulation.”
6. Line 36, 139, 313 & 324: As far as I know, shortened forms such as “it’s” are avoided for scientific writing.
7. Figure 1: Please clarify the figure legend and the main text. Does the main text start from Line 98? It is okay if this is a temporal error on this review form.
Reviewer 3 Report
The article “Alternative splicing in the regulatory circuit of plant temperature response” is a revision of recent advances in the knowledge about the molecular mechanisms regulating alternative splicing in plant genes in response to changes in ambient temperature, and about how altered messenger RNA processing is integrated in the physiological adaptation of plants to these changes. The review is clear and comprehensive, although in my opinion the first introduction section and the “pre-mRNA splicing process” section should be fused to avoid repetitive content. For example, splicing is defined in both lines 21 and 54; lines 23 and 100 describe alternative splicing; line 38 and line 104 refer to the interaction between temperature and alternative splicing.
The paragraph from lines 88-97 is actually a part of the Figure 1 legend, isn’t it?
Lines 105 and 107 explain the same concept. Is the paragraph from line 107 to line 114 actually the legend for Figure 2?
In figure 2: phosphoralation or phosphorylation?
My main concern about this manuscript is that authors offer an interesting discussion but the gap in knowledge is not enough well identified. It is not the mechanism of splicing, which is excessively described in the second section, but how alternative splicing is integrated in the response to temperature changes of plants. Two similar reviews were published recently (in 2021) and these citations are not included in this article:
Varvara Dikaya, Nabila El Arbi, Nelson Rojas-Murcia, Sarah Muniz Nardeli, Daniela Goretti, Markus Schmid. Insights into the role of alternative splicing in plant temperature response. Journal of Experimental Botany, Volume 72, Issue 21, 20 November 2021, Pages 7384–7403, https://doi.org/10.1093/jxb/erab234
Sheeba John, Justyna Jadwiga Olas, Bernd Mueller-Roeber. Regulation of alternative splicing in response to temperature variation in plants. Journal of Experimental Botany, Volume 72, Issue 18, 30 September 2021, Pages 6150–6163, https://doi.org/10.1093/jxb/erab232
These papers refer to how plant temperature response is mediated by alternative splicing, which is a part of the content of this current review. Therefore, it could still be relevant to the scientific community if authors would focus on the upstream regulation of alternative splicing and less in the downstream impact of alternative splicing.
Minor points:
Line 168: splicing factors “were” reported
Line 328: sever?
Some references do not follow the required format.
Round 2
Reviewer 1 Report
The authors have addressed all comments and suggestions, and the amended version is suitable for publishing in IJMS.
Author Response
We thank the Reviewer for their effort to improve our manuscript.
Reviewer 3 Report
The manuscript has been corrected in most of the points I suggested, but there are still some typographic and grammatical mistakes. The names of many journals are not abbreviated. I attach a modified version in pdf, with marks, to help with that.

Author Response
We apologize for the typographic and grammatical mistakes. We thank the Reviewer for their detailed reading and the effort they have made to improve our manuscript. In our resubmitted manuscript, the typographic and grammatical errors have been revised in Lines 1, 4, 11, 40, 71, 117, 137, 164, 168, 173, 188, 300, 320 and 322. We have also revised the format of journal names in Lines 418, 420, 422, 423, 428, 431, 433, 436, 448, 452, 469, 513, 523, 524, 540, 553, 555, 556, 582, 586, 598, 601, 604, 606, 608, 610, 629, 631, 639 and 641.